# Synthesis and Characterization of a Bioconjugate Based on Oleic Acid and L-Cysteine

**DOI:** 10.3390/polym13111791

**Published:** 2021-05-29

**Authors:** Marco Vizcarra-Pacheco, María Ley-Flores, Ana Mizrahim Matrecitos-Burruel, Ricardo López-Esparza, Daniel Fernández-Quiroz, Armando Lucero-Acuña, Paul Zavala-Rivera

**Affiliations:** 1Departamento de Física, Universidad de Sonora, Blvd. Luis Encinas y Rosales S/N, Colonia Centro, 83000 Hermosillo, Sonora, Mexico; marco.vizcarra@unison.mx (M.V.-P.); ricardo.lopez@unison.mx (R.L.-E.); armando.lucero@unison.mx (A.L.-A.); 2Pritzker School of Molecular Engineering, The University of Chicago, Chicago, IL 60637, USA; mleyf@uchicago.edu; 3Departamento de Investigación en Polímeros y Materiales, Universidad de Sonora, Av. Colosio S/N, Colonia Centro, 83000 Hermosillo, Sonora, Mexico; a213220985@unison.mx; 4Departamento de Ingeniería Química y Metalurgia, Universidad de Sonora, Av. Colosio S/N, Colonia Centro, 83000 Hermosillo, Sonora, Mexico; daniel.fernandez@unison.mx

**Keywords:** surfactant, oleic acid, L-cysteine, bioconjugation, biomolecule

## Abstract

One of the main challenges facing materials science today is the synthesis of new biodegradable and biocompatible materials capable of improving existing ones. This work focused on the synthesis of new biomaterials from the bioconjugation of oleic acid with L-cysteine using carbodiimide. The resulting reaction leads to amide bonds between the carboxylic acid of oleic acid and the primary amine of L-cysteine. The formation of the bioconjugate was corroborated by Fourier transform infrared spectroscopy (FTIR), Raman spectroscopy, and nuclear magnetic resonance (NMR). In these techniques, the development of new materials with marked differences with the precursors was confirmed. Furthermore, NMR has elucidated a surfactant structure, with a hydrophilic part and a hydrophobic section. Ultraviolet-visible spectroscopy (UV-Vis) was used to determine the critical micellar concentration (CMC) of the bioconjugate. Subsequently, light diffraction (DLS) was used to analyze the size of the resulting self-assembled structures. Finally, transmission electron microscopy (TEM) was obtained, where the shape and size of the self-assembled structures were appreciated.

## 1. Introduction

This work focused on the synthesis of new biomaterials from the bioconjugation of oleic acid with L-cysteine using carbodiimide. The process of chemically joining two or more molecules through a covalent bond with the help of a crosslinking agent is called bioconjugation. These agents have reactive ends for specific functional groups (primary amines, sulfhydryls, etc.) [1]. The demand for biocompatible and biodegradable materials for medical applications has increased the research of new compounds that satisfy those properties. Within these new compounds, some surfactants could meet these characteristics [2,3]. Surfactants have a characteristic molecular structure comprising a group (generally a long alkyl chain) with little attraction for the solvent (hydrophobic, if the solvent is water), together with another group (ionic or non-ionic) that shows a strong attraction of the solvent (hydrophilic in aqueous systems). The degree of surface activity and the type of application depends on the hydrophilic/lipophilic balance (HLB) characteristics of this amphiphilic structure [4]. The surfactant molecules of renewable raw materials that mimic natural lipoamino acids are one of the preferred options for food, pharmaceutical, and cosmetic applications [5]. These surfactants can be obtained from molecules that mimic natural amphiphilic structures. Based on these considerations, and as part of our study objective, a surfactant was synthesized from oleic acid and cysteine (lipoamino acids). The association of polar amino acid and a non-polar long-chain compound to build amphiphilic structures allows obtaining molecules with a high surface activity [6]. When an amphiphilic molecule is dissolved, the lyophobic group distorts the structure of the solvent, increasing the free energy in the system [7]. The presence of an amino acid as a polar group in an amphiphilic molecule characterizes this type of surfactant. For its synthesis, amino acids used as protein building units, such as natural materials, are used. With the amino acids having a carboxylic group and a primary amino group, surfactants synthesized from them can be prepared by adding the hydrophobic component in either of the two parts, as presented in the literature [3].

Some studies indicate that amino acid-based surfactants form fibrillar structures are capable of gelling oils [8]. The nature of these fibers lies in the chiral self-assembly of amphiphilic molecules. The fibers form a three-dimensional network through the medium to form a gel. In general, the materials that exhibit these properties are referred to as low molecular weight gelling agents [9].

The amino acid can be derivatized by reacting to its carboxylate group with an alcohol to produce an aminated ester or with an amine to produce an amino amide. Alternatively, the amino group can be started by reacting it with a fatty acid to obtain an acidic amide or with an alkyl halide to form a secondary or tertiary amine, thus, producing an N-akylaminoacid [2,3,5]. The term lipoamino acids apply to those compounds in which a single amino acid is linked, either through its carboxyl or amino group, to a long-chain fatty acid [10].

In 1909, S. Bondi synthesized lipoamino acids from glycine and alanine as N-lauroyl amino acids to elucidate the theory that fat necrosis in cells was attributed to the breakdown of acylamino acids into amino acids and fatty acids [11]. Later, in 1955, the natural existence of lipoamino acids was confirmed [12]. Since then, several lipoamino acids have been found in different living things [10].

In this work, L-cysteine was chosen because it contains a thiol group (-SH). The advantage of containing a thiol group is that sulfur in its different oxidation states represents one of the most versatile elements in molecular biology. Many of the sulfur compounds in living things, such as some vitamins, methionine, and cysteine, are essential for human nutrition by carrying out functions such as electron transport in Fe-S centers and structural roles through disulfide bridges [13]. Disulfide bridges in proteins improve the thermodynamic stability of the molecule and can be reversible in reducing environments. The reactivity of the individual cysteine thiols can be used to manipulate the oxidative assembly reactions [14,15]. These molecules are self-assembled due to London, Van der Waals dispersion forces, hydrogen bonds, and electrostatic and interfacial attraction and repulsion forces [8]. Besides, thiol groups are well known in nanotechnology for their property for self-assembly or self-organizing on specific metallic surfaces, like gold, silver, copper, among others.

The bioconjugation process involves the reaction between a carboxylic acid such as oleic acid with the amino group of an amino acid such as L-cysteine, forming an amide bond. However, the ideal method of amide synthesis by direct condensation of a carboxylic acid and an amine for the formation of an equivalent molecule of water as the sole by-product is not practical. As a result, activation of the acid component is necessary to promote amide coupling and develop an efficient synthesis process [16]. Like an alternative is used, the carbodiimide mediated conjugation, reported in 1955 [17]. For this type of conjugation, it is important to select carbodiimide based on the solvent, considering that a separation of the corresponding urea must occur. The case of reactions that start from fatty acids or a compound that is also soluble in non-polar or amphiphilic solvents, such as DMSO, should be used, and that the corresponding urea can be easily removed.

## 2. Materials and Methods

### 2.1. Materials

Oleic acid (natural, FCC), N-hydroxysuccinimide (98%), sodium hydroxide (ACS), dimethylsulfoxide (ACS), and hydrochloric acid (36.5–38%) were obtained from Sigma-Aldrich, St. Louis, MO, USA. L-cysteine (high purity grade) was obtained from Amresco, Pennsylvania, N,N′-dicyclohexylcarbodiimide of Alfa Aesar, Tewksbury, Massachusetts,). All materials were used without further modifications.

### 2.2. Methods

Preparation of amphiphilic molecules: For the formation of the final bioconjugate, a two-step synthesis system mediated by N′-dicyclohexylcarbodiimide (DCC) and N-hydroxysuccinimide (NHS) was carried out. This step corresponds to the formation of a stable ester from oleic acid by coupling carbodiimide with the carboxyl group of the acid, followed by its replacement by NHS. The next step corresponds to the nucleophilic attack of the primary amine of L-cysteine. The N-hydroxysuccinimide esters formed during the first stage have sufficient stability to allow a two-step reaction in which the conditions of the solution can be adjusted first to favor the formation of esters and subsequently adjusted to favor the amidation reaction [18], as show in Figure 1**.** The solvent used for the synthesis was DMSO.

Esterification of the oleic acid: 5 mL of a 0.002 M solution of Oleic acid, 5 mL of a 0.002 M solution of dicyclohexylcarbodiimide, and 5 mL of a 0.004 M solution of N-hydroxysuccinimide were prepared using a polar aprotic solvent such as dimethyl sulfoxide as the medium of reaction. For the first step of the synthesis of lipoamino acid, the solutions of freshly prepared N-hydroxysuccinimide and diciclohexylcarbodiimide were mixed in 5 mL of solvent. Due to the instability of O-acylisourea, the temperature decrease is suggested to increase the half-life of said molecule in solution [18]. For this reason, the solution was subjected to magnetic stirring in a cold bath at 10 °C to prevent the formation of N-oleylurea once the oleic acid is added. During this step, the protonation of the carbodiimide nitrogen facilitates the formation of the O-acylisourea because said protonation reduces the electron density in the central carbon of the carbodiimide, favoring the nucleophilic attack of the carbonyl group as shown in the literature [19]. Taking into account the above and considering that oleic acid has a pKa of 5.02, the solution of oleic acid was added, and the pH was lowered to 4 using a 0.1 M hydrochloric acid solution to promote the protonation necessary to the formation of the NHS-ester. In this step, the precipitation of dicichlohexylurea occurs as N-hydroxysuccinimide replaces the carbodiimide coupling, as reported in the literature.

Amidation with L-cysteine: The reaction of the N-hydroxysuccinimide ester with an amine is an example of a nucleophilic attack, for which the amine must remain deprotonated. After 15 min of stirring, the solution was adjusted to pH 9 using a solution of 1 M sodium hydroxide, and 0.0006 moles of L-cysteine was added directly. The reaction was left under stirring for 3 h and subsequently stored in a conical tube to precipitate the by-product corresponding to dicyclohexylurea and N-hydroxysuccinimide completely.

Separation of reaction products: The mixture was left to stand for 24 h, to induce the separation of the non-soluble undesired residues (mostly urea subproducts) by gravity. The supernatant was extracted using a micropipette and stored for further purification. A 4 mL sample was taken from the homogeneous solution to which 1 mL of a 0.3 M sodium hydroxide solution was added until a pH of 10 was reached to induce the formation of a gel; characteristic of amino acid-based surfactants [8]. The sample was left to rest for 24 h until complete gelification, then proceed to their separation from the rest of the reactant and residues left in liquid form. The route consisted of the direct extraction of the gel using tweezers, followed by two washes with deionized water, freeze-dried, and crushing the product with an agate mortar (Figure 2).

### 2.3. Characterization Techniques

The dry form obtained by gelation was used for the characterization techniques herein described. High-resolution ^1^H-NMR spectroscopy was recorded on a Bruker Avance 400 spectrometer operating at 400 MHz. The sample was solubilized in DMSO-d6 (99.9 atom % D) using TMS as an internal reference. For FTIR spectroscopy, the spectral data were obtained using a Thermo Scientific brand spectrometer model Nicolet iS50, the samples were dry and placed directly on the diamond window of the ATR mode (attenuated total reflectance), and the spectrum was measured in a range between 4000 and 500 cm^−1^ (64 scans per sample). For Raman spectroscopy, experiments were performed at room temperature using a 473 nm diode laser (Excelsior One 473, Spectra-Physics, MKS Instruments, Inc, Santa Clara, California, CA, USA) with an incident power of about 5.0 mW. The light was focused and collected with a 100× objective. The emitted light was detected with a 0.55 m spectrometer (iHR550, Horiba, Miyanohigashi, Japan) and a cooled CCD (CCD-1024x256-BD-SYN, Horiba). The elastic component of the collected light was removed by a Notch filter. DLS experiments were done using a Brookhaven BI-200SM Research Goniometer System with a 637 nm He–Ne laser and 15 mW power. The intensity correlation function was measured at a fixed angle of 90 degrees. UV-visible spectrophotometry was performed with the VWR brand double beam spectrophotometer; model UV-6300PC with UV-Visible scanning. This technique was used to determine the critical micellar concentration (CMC) value roughly, whereas a micellar solution acts as a colloidal suspension relative to the wavelength of light [20]. This method is based on the change in absorption behavior after aggregation [21]. For transmission electron microscopy (TEM), a JEOL JEM-2010F field emission apparatus (Akishima-shi, Japan) was used, operating at 200 keV on a 200-mesh carbon on a copper grid (Ted Pella) followed by a drop of 3% phosphotungstic acid as a staining agent. Micrographs were taken and analyzed using Gatan software (Gatan Inc., Pleasanton, CA, USA).

## 3. Results

Nuclear magnetic resonance characterization: The ^1^H-NMR spectrum of *N*-oleyl-L-cysteine bioconjugate (MAT) is presented in Figure 3. The signal corresponding to the methyl protons was found at δH = 0.85 (a), followed by the signals corresponding to the alkyl protons CH_2_ of the oleic acid chain δH = 1.29–1.20 (b). The protons of the methylene group were found in δH = 5.32 (d), and the most immediate carbon protons to this were found at δH = 1.97 (c). Because deuterated DMSO was used as a solvent for this experiment, a representative substitution of the hydroxyl group and amide group protons was not observed. Therefore, the proton signal of the hydroxyl group was found in δH = 10.35 (h) and δH = 6.96–7.22 (g) the signals of the protons corresponding to the three amide groups found in the synthesized molecule were found [22]. The characteristic proton signals and integrals were suggestive of bioconjugate formation, and the proposed structure is shown inside (Figure 3).

The spectrum presented in Figure 3 corresponds to the purified bioconjugate, but the signals of some reaction impurities were observed. In this sense, the signals at δH = 1.58–1.73 are associated with the alkyl protons of the cyclohexyl rings of the *N*-acyl urea, which is a secondary reaction product. Similarly, it is identified at δH = 5.59, the nitrogen proton in the urea group [23].

Infrared spectroscopy: In the spectrum corresponding to oleic acid (OA). We can observe in 3005 cm^−1^ the signal corresponding to the stretching of the C–H bond adjacent to its unsaturation. We also found two very marked bands in 2921 and 2852 cm^−1^, corresponding to the asymmetric and symmetrical stretches, respectively, of C–H of the alkyl chain. In 1707 cm^−1^, was found the characteristic band of the carbonyl group present in said fatty acid corresponding to the C=O stretching of the carbonyl. Moreover, in the spectrum for L-cysteine (CYS), we observe in 3163 cm^−1^ the representative band of the N-H stretch of the amino group, the characteristic vibrations of the thiol group S-H can be observed in 2550 cm^−1^, and the C-S link appears in the 635 cm^−1^ [15,24,25]. Finally, the spectrum for the oleic acid and L-cysteine bioconjugate (MAT) was obtained from a solid-state sample. Comparing the spectrum of the bioconjugate with that of L-cysteine, the following bands were found. First, at 3320 cm^−1,^ the first difference for the spectrum compared to the L-cysteine spectrum, which corresponds to the N-H stretching vibration of the amide group. At 2580 cm^−1^, the associated band of the S-H from the thiol group was found shifted from the earlier band at 2543 cm^−1^ present in the molecule. The carbonyl group C=O is observed in 1705 cm^−1^. In 1624 cm^−1^, a new band corresponding to the vibration of the carbonyl group set called amide appears, In addition, carboxylic acids show characteristic C–O stretching in 1240 cm^-1^. Besides, the band corresponding to the C–S link is pronounced in 643 cm^−1^, indicating a strong presence of this link in the product formed [24,25,26,27] (Figure 4).

The Raman spectroscopy: The spectrum of L-cysteine shows signals associated with its chemical structure, in the region of 200 to 500 cm^−1,^ the ones associated with the vibrations of CCC bend, CCC rock, CCN bend, and CCS bend [28]. Moreover, the signal at 640 cm^−1^ corresponds to the stretching vibrations of the CS link. The peak observed in 2544 cm^−1^ corresponds to the HS link of mercaptan [29]. Regarding the Raman spectroscopy of oleic acid, the literature establishes the peaks of 1000–1200 and 2800–3000 cm^−1^, which are indicators of the torsion and packing signal, respectively, of the chain. As well as signals in the region of 2850 and 2930 cm^−1^ corresponding to the vibrational modes of asymmetric CH_2_ and symmetric CH_3_, respectively, and sensitive to hydrophobic interactions between hydrocarbon chains and indicate chain packing [30].

In the spectrum analysis, we found in 720 cm^−1^ the signal of the thiol SH group, corresponding to a stretch in phase; later, we observed the signal of the CH_2_ group corresponding to the aliphatic chain of oleic acid in 731 cm^−1^ of the rock movement in phase; in 1417 cm^−1^ the CH signal appears, the C–NH link of stretching and bending appears in 1551 cm^−1^, the double bond of carbon–carbon we see in 1660 cm^−1^, assigned to stretching, the carbonyl group appears in 1751 cm^−1^, assigned to stretching in phase. The signal of the NH group of the amide is observed at 2327 cm^−1^, which confirms the presence of the peptide bond. Subsequently, a broad band appears, in which some signals such as the fermi resonance of CH_2_ in 2890 cm^−1^ are combined, the overtone of the C–NH bond in 2943 cm^−1^, and the signal of a CH_3_, of the oleic acid chain. Furthermore, an O–H stretch band around 3428 cm^−1^ in the Raman spectrum of (MAT) is indicative of carboxylic acid groups on L-cysteine [29] (Figure 5).

Critical micellar concentration (CMC): The critical micellar concentration of a surfactant solution can be easily determined by plotting the absorbance as a function of surfactant concentration. The cross intersection between the two slopes is taken as the CMC of that solution. The interaction between solute–solute molecules and solute–solvent molecules can give useful information about the interfacial and thermodynamic properties. These behaviors are attributed to the delicate balance between the hydrophobic and hydrophilic interaction of the surfactant solution [20]. The different concentrations of the surfactant were analyzed using a quartz cuvette at different concentrations of MAT in deionized water.

By applying the UV-Vis technique, the CMC of a new material was determined, a solution with 0.0012 g of MAT in solid state was prepared and dissolved in 0.002 L of deionized water, which is equivalent to a concentration of 0.600 g/L. This concentration marked the beginning of UV-Vis measurements (Figure 6). The wavelength used was 204 nm, which corresponds to the length at which cysteine is absorbed. The value determined for CMC = 0.226 g/L. This value is considered approximate to the actual value of CMC.

The CMC was determined by plotting the absorbance against concentration. The graph presents two parts with different slopes, and the CMC was evaluated from their point of intersection. The CMC values obtained from the given UV-Visible data are shown in Figure 7.

Dynamic scattering of light (DLS): To determine the hydrodynamic ratio of the particle, dynamic light scattering experiments (DLS) were performed. A dynamic light scattering experiment consists of having a laser beam effect sample and collecting the light scattered at an angle *θ* with a photomultiplier. The intensity fluctuations of the scattered light allow obtaining the normalized correlation function.
$$g_{2}\left(\tau \right)-1=B{\left|g_{1}\left(\tau \right)\right|}^{2}$$
where *τ* is the characteristic time of the decay of the correlation and *B* is a parameter that depends on the experimental geometry. In the case of a monodisperse solution, the field correlation function decays as $g_{1}\left(\tau \right)=\exp\left(-\Gamma t\right)$ with a decay given by $\Gamma =Dq^{2}$ where *D* is the mutual diffusion coefficient and $q=\frac{4\pi n}{\lambda }\sin\left(\frac{\theta }{2}\right)$ is the magnitude of the scattering vector, *n* the refraction index, *λ* the wavelength, and *θ* the scattering angle.

Two samples were measured, the first one at a CMC concentration obtained by UV-Vis (0.226 g/L) as shown in Figure 7; only noise was detected, and a hydrodynamic radius could not be established. The second sample was used at a concentration above the CMC calculated at 0.3 g/L.

The sample cells were 2-mL cylindrical ampules (concentration = 0.3 g/L) immersed in an index-matching bath of decline, thermostat at 25 °C. The sample was filtered and sonicated for 10 min, before measurements. Γ was obtained by adjusting the correlation function for a decaying exponential and was used in the Stoke–Einstein equation.
$$D=\frac{k_{B}T}{6\pi \eta R}$$
where *R* is the hydrodynamic radius, *k_B_* is Boltzmann’s constant, *T* is the temperature, and *η* is the viscosity of the solvent. The value obtained for the hydrodynamic radius is 50.57 nm, and the polydispersity index (PDI) is 0.1602 (Figure 8 and Figure 9).

Transmission Electron Microscopy (TEM): The morphology observed in the micrographs shows spherical and cylindrical shapes (Figure 10). The average particle radius value found is 31.81 nm obtained using ImageJ 1.51m9 software from Wayne Rasband, National Institute of Health (500 particles analyzed), presented in Figure 11.

The values TEM using a JEOL JEM-2010F Field Emission apparatus (Akishima-shi, Japan) operating at 200 keV on a 200-mesh carbon on a copper grid (Ted Pella) followed by a drop of 3% phosphotungstic acid staining agent. Micrographs were taken and analyzed using Gatan software (Gatan Inc., Pleasanton, CA, USA).

## 4. Conclusions

In this work, we sought to obtain a bioconjugate of L-cysteine and oleic acid through a coupling reaction assisted by carbodiimide. ^1^H-NMR spectroscopy was employed to elucidate a molecule of the bioconjugate material; the signals and integrals characteristics of the protons suggested the formation of a molecule formed by one oleic acid molecule and three molecules of cysteine. Infrared spectroscopy confirmed the formation of an amide group that resulted from the coupling reaction and suggested that for the synthesis conditions used, there were no disulfide bridges. An approximate value of the micellar critical concentration (CMC) was obtained using UV-Vis and TEM confirms self-assembling structures of spherical and cylindrical shapes with an average size radius of 31.81 nm; comparable to the 50.57 nm of hydrodynamic radius obtained by DLS. In addition, the study is underway to explore the possible applications, within which we could include that of encapsulant of bioactive molecules with hydrophobic properties or as surface modification of gold, silver, or any other metal, reactive to thiol groups nanoparticles.

## 5. Patents

This research has been submitted for a licensed patent in Mexico with the file number MX/a/2018/016047, dated 12 December 2018.

## Figures and Tables

**Figure 1 polymers-13-01791-f001:**
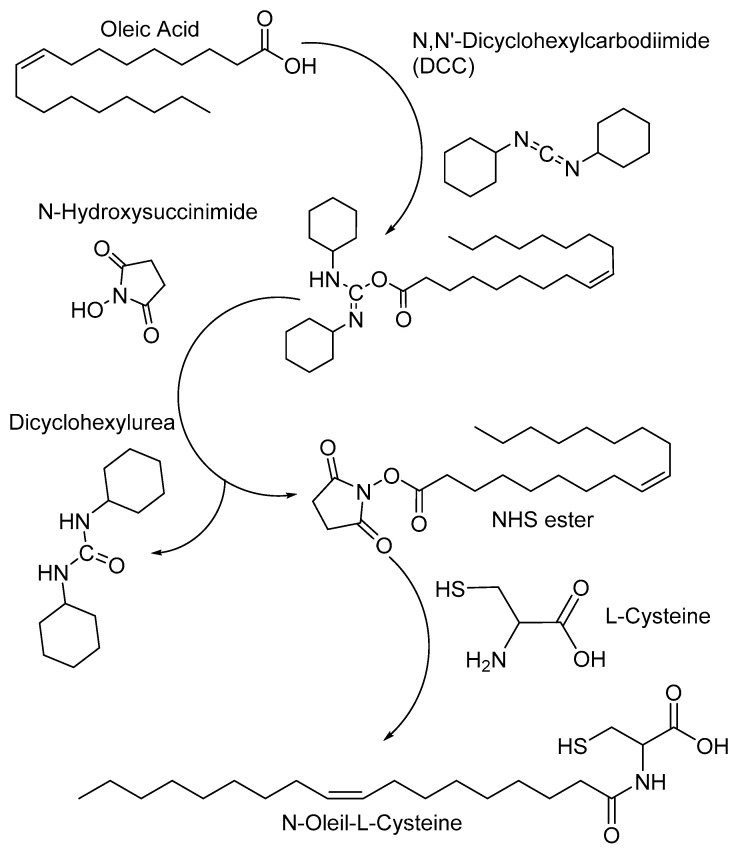
General scheme of the carbodiimide-based conjugation reaction for the synthesis of amphiphilic molecules of N-oleyl-L-cysteine.

**Figure 2 polymers-13-01791-f002:**
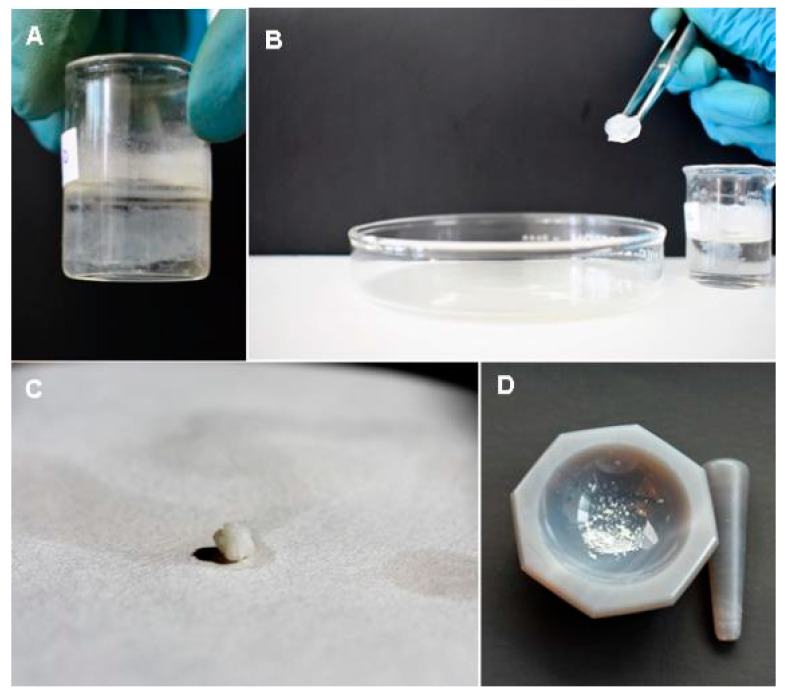
Process of separation of the bioconjugate of oleic acid and L. cysteine: (**A**) Formation of a gel at pH 12, (**B**) Physical extraction of the gel, (**C**) Drying, and (**D**) Crushing of the sample.

**Figure 3 polymers-13-01791-f003:**
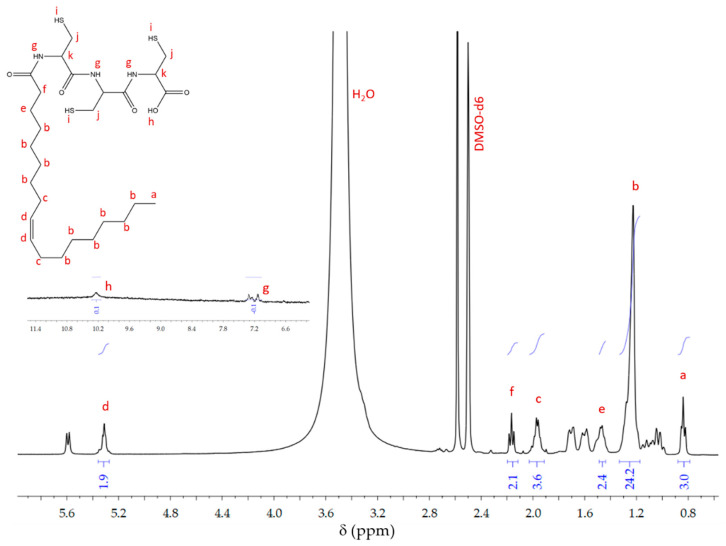
^1^H-NMR spectrum of (MAT).

**Figure 4 polymers-13-01791-f004:**
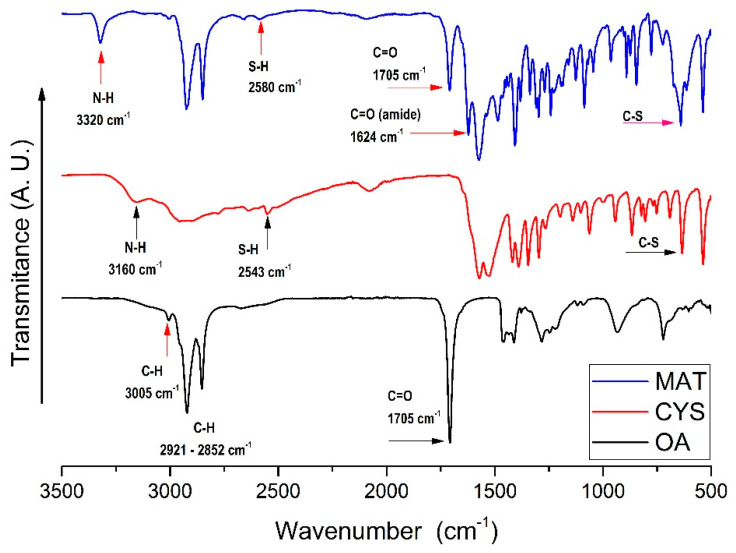
Infrared spectra; Oleic acid (**OA**), L-cysteine (**CYS**), and (C) Bioconjugate of oleic acid and L-cysteine (**MAT**).

**Figure 5 polymers-13-01791-f005:**
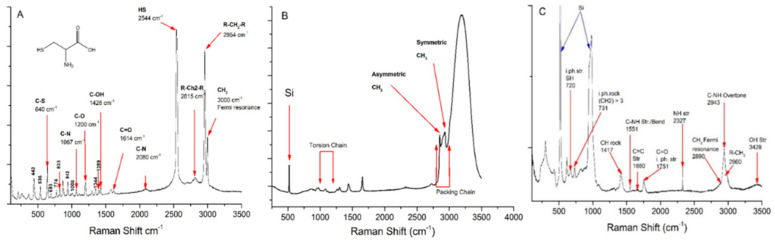
Raman spectrums. (**A**) Amino acid L-cysteine. (**B**) Oleic acid, (**C**) bioconjugated material of oleic acid and L-cysteine (MAT).

**Figure 6 polymers-13-01791-f006:**
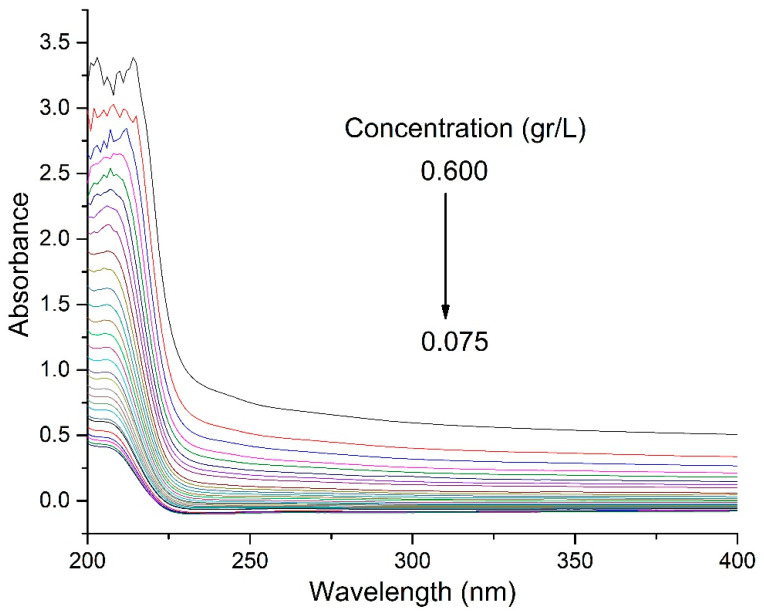
UV-Vis spectrum of bioconjugate of oleic acid and L-cysteine (MAT).

**Figure 7 polymers-13-01791-f007:**
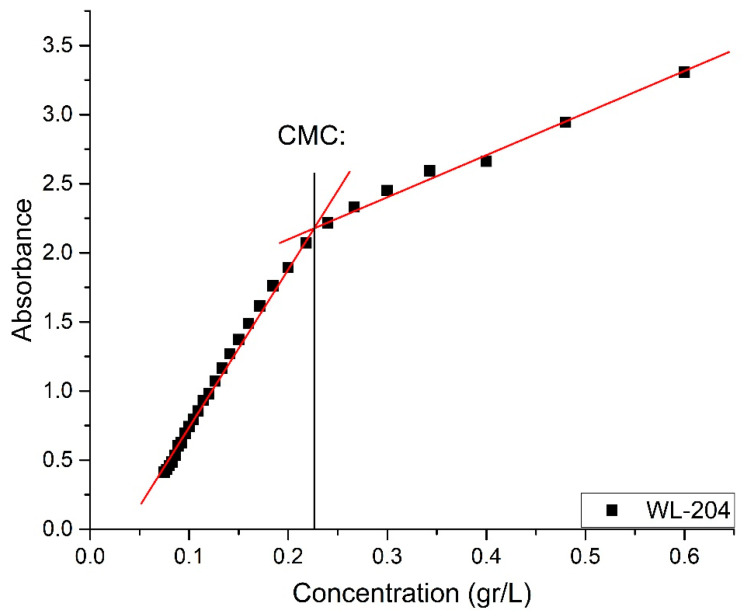
Absorbance versus concentration plots of bioconjugate of oleic acid and L-cysteine (MAT). Wavelength 204 nm.

**Figure 8 polymers-13-01791-f008:**
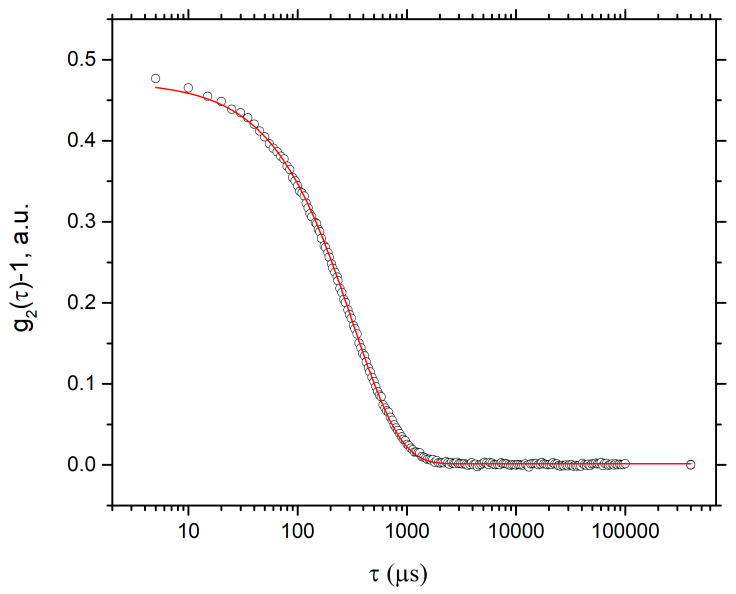
Correlation curve obtained by DLS of the MAT sample at 0.3 g/L at 25 °C.

**Figure 9 polymers-13-01791-f009:**
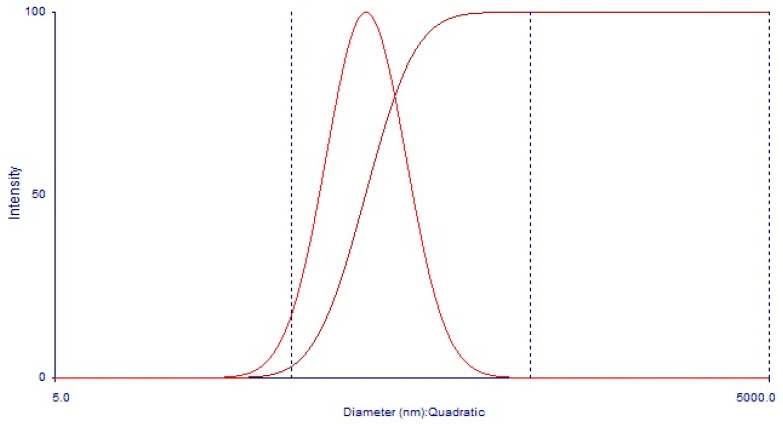
Hydrodynamic diameter distribution of 101.14 nm of the MAT sample at 0.3 g/L by DLS.

**Figure 10 polymers-13-01791-f010:**
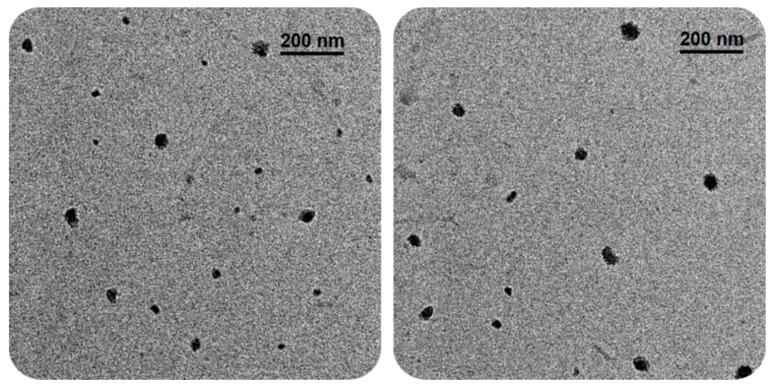
TEM images show spherical and cylindrical particles and sizes less than 100 nm.

**Figure 11 polymers-13-01791-f011:**
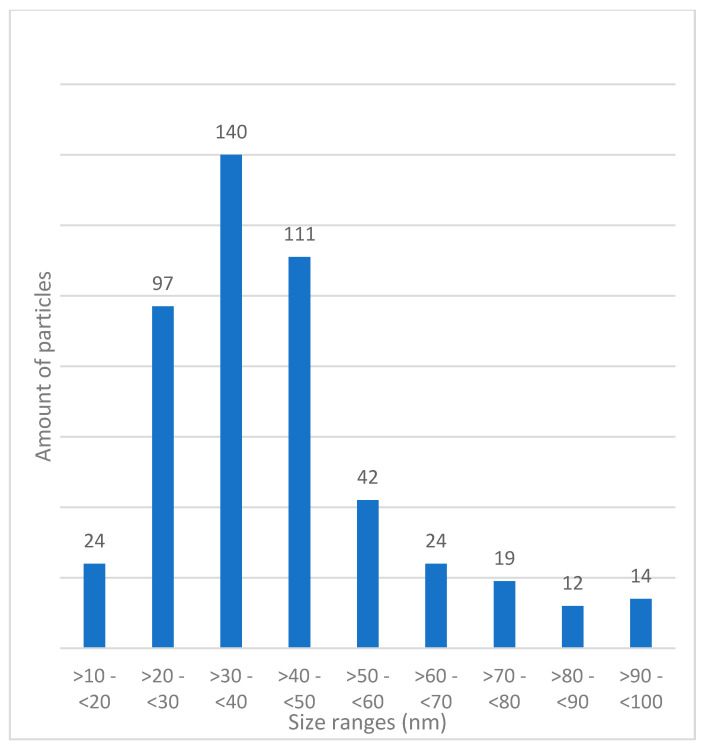
Histogram of sizes obtained in TEM. On the X-axis, we have size ranges (nm), and on the Y-axis, we have the number of particles.

## Data Availability

The data presented in this study are available on request from the corresponding author.

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
