# Peer review of "Synthesis and Characterization of a Bioconjugate Based on Oleic Acid and L-Cysteine"

_polymers, 2021, doi:10.3390/polym13111791_

Round 1

Reviewer 1 Report

In this article authors report on the synthesis and characterisation of a proposed "novel "oleic acid covalently linked to L-cysteine. Although the characterisation is presented the application is not clear and may not directly fit in the scope of the journal.

Comments

  1. There are instances (line 147, 183, 214, 226, 263, 275 ) where references are missing" (Error! Reference source not found.) "
  2. Elaborate on the stability of the proposed bioconjugate.
  3. Please check English translations for e.g "acquiral ".
  4. In addition to 1H NMR, 13C NMR should also be carried out.  Furthermore peak integration (number of equivalent protons) in the 
  5. 1H NMR should be included in the manuscript.
  6. Please elaborate on " This material showed the potential 19
    to be used in the encapsulation of active compounds for drug delivery applications. "

Author Response

First of all, I would like to thank you on the time you take to review our work, herein I add the changes made on the work due tu your recommendations.

  1. Thanks, we are having troubles with Mendeley, we fixed the links, hopefully the will work in your computer as well.
  2. Ready, we expand the reaction with the proper literature.
  3. We change it
  4. We expanded on the 1H NMR results, sadly we were unable to measured through 13C due to the lack of liquid He and its difficulty to get some on pandemic days. But, we hope that the next month the tanks will be refill.
  5. 1H NMR is included.
  6. Done

Reviewer 2 Report

The present manuscript requires a substantial revision and reorganization to be suitable for publication.

Introduction: It is not clear the aim of the manuscript. Specifically, it is not clear if the authors have planned to synthesize polycysteine-based oleoyl surfactants or not, and, in general,  which is the focus of the manuscript. For this reason, the introduction is rather vague and must be re-written in a more focused way.

Methods: The methods are not sufficient. In the method section, it was reported only the method for the synthesis and not for the instrumental techniques employed. The separation and purification methods proposed are quite uncommon. Are they reproducible? Are the authors convinced that they are effective in the purification of the product? Why the product form a gel at basic pH values?

Results: The authors should present firstly the identification and characterization of the obtained compound at the solid state, then, the analysis in solution starting from CMC determination, and the, self-assembling (DLS and TEM). The method used for CMC determination is not clear and should be detailed. Possibily, another standard technique for CMC determination should be also performed. The results about CMC determination and self-assembling must be detailed and discussed in a sufficient way. Is DLS trace and TEM images referred to surfactant micelles or other aggregates form? The presented information in the manuscript are too generic.

The authors have to discuss the presented results with the literatures both as regard the synthetic approach and the chemical-physical characterization.

Conclusions: Conclusions have to be modified accordingly and improved.

There are several typos and language mistakes.

Author Response

First, I would like to thank you for the time taken to review our work. Herein I add the changes made due to your recommendations.

  1. We improve the introduction and focus more on the text.
  2. We add more description of the techniques and justify the gelation based on literature.
  3.  The results have been modified as you recommended and typos found have been fixed.

Round 2

Reviewer 1 Report

A complete check for grammar and presentation of spectroscopic methods and analysis is required. In this regard I have highlighted a few of these concerns below and provided some examples how to report on these methods, however it is recommended that authors perform the necessary checks to clarify sentences throughout the article to ensure a good read.

Kindly address the following comments 

  1. Spelling line 140 (diciclohexylcarbodiimide), line 150 (O-oleilisourea) and Figure 1. correct spelling of N-Oleil-L-cysteine.
  2. Resolution should be enhanced for Figure 1,2 and 10.  Some text and labels are difficult to read. Although the authors have included the proton integrals the signals are very weak.
  3. Figure 4. I have several comments in regards to a) it may be easier to refer to spectra in discussion if authros label the spectra Figure 4a, b,c, b) %Transmittance should be used to label the y axis and c) “y” prefix to 2852 in Figure 4 should be removed.
  4. Line 193 Check sentence meaning “This shows at 3320 cm-1 the first difference for the spectrum of oleic acid, which corresponds to the N-H stretch vibration of the amide group”. Which spectra is being compared? 
  5. Line 224 Rephrase “a thick signal appears,” The signals for molecular vibrations are either described as broad, narrow  or sharp for example .
  6. Line 226  “And finally, we found an OH band at 3428 cm-1 [28]. (Figure 5)” It is best to report in the third person. For e.g.  A O-H stretching band around 3428 cm-1 in the FTIR spectrum of (MAT) was indicative of carboxylic acid groups in L-cysteine backbone.
  7. Line 243 Correct grammar “From this solution, diluted concentration of the surfactant were obtained and it is observed how the absorption signal decreases”
  8. Line 274 “From the observing signals in the spectrum of the reaction”- this should say of the isolated bioconjugate (MAT) for clarity?
  9. Figure 10 caption needs to be corrected “Figure 10. Nuclear magnetic resonance spectrum. ” to. 1H NMR spectrum of ____
  10. Line 283 1H NMR is not referred to as “magnetic resonance analysis”.
  11. Line 288 “. The “signs” in δH = 1.58-1.73 are associated with the alkyl protons of the cyclohexyl ring  s of the secondary reaction product, N-acyl urea”. requires rephrasing for e.g. The signals at δH = 1.58-1.73. Correct in all instances.
  12. Line 289 “The signs in δH = 1.58-1.73 are associated with the alkyl protons of the cyclohexyl ring  s of the secondary reaction product, N-acyl urea. Similarly, it is identified in δH = 5.59 the nitrogen proton in the urea group [31].” Please clarify what is the nature of the sample that was analysed by 1HNMR i.e. crude reaction mixture or purified bioconjugate (MAT)? and then need to state that these are reaction impurities.
  13. Line 311: “Infrared spectroscopy confirmed the formation of an amide group characteristic of the coupling and suggests the absence of disulfide bridges for the synthesis conditions used. “ should read better for e.g
  14. Infrared spectroscopy confirmed the formation of an amide group that resulted from the the coupling reaction and suggested that for the synthesis conditions used there were no disulfide bridges indicative of ____.
  15. Line 319 “The nuclear magnetic resonance technique confirms the coupling of three cysteine units to the oleic chain” This sentence needs to be improved on. For e.g.                              
  • 1H NMR spectroscopy was employed to elucidate/ characterise ....
  • The characteristic proton signals and integrals were suggestive of bioconjugate formation.

Author Response

Thanks a lot for your time an effort to correct the paper!

A wide range of the changes from grammar and scientific editing were made. Furthermore, changes in the resolution of the pictures were improved, according to your recommendations.

Reviewer 2 Report

The authors have answered in a not detailed way and not point-by-point to the comments of the reviewer and they did not track the changes made in the revised version of the manuscript.

Substantial revisions of the manuscript are still needed.

Reviewer (Round 1): Introduction: It is not clear the aim of the manuscript. Specifically, it is not clear if the authors have planned to synthesize polycysteine-based oleoyl surfactants or not, and, in general, which is the focus of the manuscript. For this reason, the introduction is rather vague and must be re-written in a more focused way.

Authors: We improve the introduction and focus more on the text.

Rewiever (Round 2). The introduction is still vague and it is not clear if the authors aimed to synthesize a N-oleyl monocysteine or a N-oleyl polycysteine (specifically an oleyl-based surfactant with three cysteine residues, as resulted from NMR analyses. From the introduction, I undestand the rationale for a generic surfactant containing cysteine but not if there is any relevance for synthesizing a polycysteine surfactant.

Reviewer: Methods: The methods are not sufficient. In the method section, it was reported only the method for the synthesis and not for the instrumental techniques employed. The separation and purification methods proposed are quite uncommon. Are they reproducible? Are the authors convinced that they are effective in the purification of the product? Why the product form a gel at basic pH values?

Reviewer (Round 2) The authors have added an experimental section containing the methods for the instrumental techniques used. This section should be moved after the paragraphs reporting the materials and the description of the synthetic procedure. The purification procedure of surfactant is still unclear. Were the two options proposed for the purification (extraction of the gel using tweezers and dyalisis) performed consecutively or independently? If independently, the authors should report and discuss the purity from both methods. Otherwise, they must indicate which is the final purification option chosen for the final product. The indicated reference [10] supports that, as known, this class of surfactants can be used as organogelators at some conditions, but it does not support the gelation as a method for surfactant purification. Please discuss this aspect and the method of purification.  

Reviewer: Results: The authors should present firstly the identification and characterization of the obtained compound at the solid state, then, the analysis in solution starting from CMC determination, and the, self-assembling (DLS and TEM). The method used for CMC determination is not clear and should be detailed. Possibily, another standard technique for CMC determination should be also performed. The results about CMC determination and self-assembling must be detailed and discussed in a sufficient way. Is DLS trace and TEM images referred to surfactant micelles or other aggregates form? The presented information in the manuscript are too generic.

Reviewer (Round 2) Nuclear magnetic resonance spectroscopy paragraph should be moved at the beginning of the results section, followed by FT-IR and Raman, since these techniques allowed the identification and characterization of the synthesized compound at a solid state. The method of CMC determination should be detailed in the method section and not reported in the results section. How the experimental data were fitted to find out the two lines from which CMC was calculated, it should be also reported. I am not convinced about the UV-vis method for CMC determination of this surfactant since the compound has not a cromophore in its structure and 204 nm is a low and aspecific wavelenght. I would be better to use another common technique to determine CMC (tensiometry, fluorescence usinf pyrene as probe). If it is not possible, please support the use of UV-vis for CMC determination according to the literature available. As regard DLS, was the data from the correlation function good for the analysed concentration above CMC? The reported size is quite large for common surfactant micelles. It would be better referring to surfactant aggregates than micelles. Moreover, the size for these aggregates is 31.81 nm, while from DLS the size is 214.7 nm. Generally, particle size from DLS and TEM are not identical, but there is a too large discrepancy in the presented results. Line 305-308 refer to the method used and are not pertinent in the results section.

Reviewer (Round 1): The authors have to discuss the presented results with the literatures both as regard the synthetic approach and the chemical-physical characterization.

Reviewer (Round 2) No discussion was added by the authors in reference to the available literature for N-oleyl cysteine derivatives.

Reviewer (Round 1) Conclusions: Conclusions have to be modified accordingly and improved.

The conclusions have been modified by authors but there are statements not supported by the presented results.

Reviewer (Round 2) Line 316-318 “These differences could be explained due to the effect of the pH and temperature during the measurements and the obtention of multilayer micelles at higher concentrations” The temperature and pH were not changed during the measurements. Moreover, what the authors mean for “multilayer micelles”? How these aggregates was observed?

Line 318-319 “TEM confirm that spherical self-assembly structures occur under dissolution conditions with an average size of 31.81 nm when dry.” In the results section, It was stated that the observed aggregates were spherical and cylindrical.

Reviewer (Round 1) There are several typos and language mistakes.

Reviewer (Round 2) Language mistakes and typos are still present

Author Response

First, I would like to thank you again, for your time and effort of making this work better and suitable for publication.

Rewiever (Round 2). The introduction is still vague and it is not clear if the authors aimed to synthesize a N-oleyl monocysteine or a N-oleyl polycysteine (specifically an oleyl-based surfactant with three cysteine residues, as resulted from NMR analyses. From the introduction, I undestand the rationale for a generic surfactant containing cysteine but not if there is any relevance for synthesizing a polycysteine surfactant.

Response: We try to improve and clarify more deeply the objetives in the introduction; as well, due this we decide to modified the title of the research according with this objectives.

Reviewer (Round 2) The authors have added an experimental section containing the methods for the instrumental techniques used. This section should be moved after the paragraphs reporting the materials and the description of the synthetic procedure. The purification procedure of surfactant is still unclear. Were the two options proposed for the purification (extraction of the gel using tweezers and dyalisis) performed consecutively or independently? If independently, the authors should report and discuss the purity from both methods. Otherwise, they must indicate which is the final purification option chosen for the final product. The indicated reference [10] supports that, as known, this class of surfactants can be used as organogelators at some conditions, but it does not support the gelation as a method for surfactant purification. Please discuss this aspect and the method of purification.  

Response: According to your recommendation, the texts from characterization and synthesis were moved, furthermore the purification methodology was change and the different steps were broad, explaining why they were used.

Reviewer (Round 2) Nuclear magnetic resonance spectroscopy paragraph should be moved at the beginning of the results section, followed by FT-IR and Raman, since these techniques allowed the identification and characterization of the synthesized compound at a solid state. The method of CMC determination should be detailed in the method section and not reported in the results section. How the experimental data were fitted to find out the two lines from which CMC was calculated, it should be also reported. I am not convinced about the UV-vis method for CMC determination of this surfactant since the compound has not a cromophore in its structure and 204 nm is a low and aspecific wavelenght. I would be better to use another common technique to determine CMC (tensiometry, fluorescence usinf pyrene as probe). If it is not possible, please support the use of UV-vis for CMC determination according to the literature available. As regard DLS, was the data from the correlation function good for the analysed concentration above CMC? The reported size is quite large for common surfactant micelles. It would be better referring to surfactant aggregates than micelles. Moreover, the size for these aggregates is 31.81 nm, while from DLS the size is 214.7 nm. Generally, particle size from DLS and TEM are not identical, but there is a too large discrepancy in the presented results. Line 305-308 refer to the method used and are not pertinent in the results section.

Response: the change in the results presented in the text, were moved. The CMC methodology was extended and supported by the literature. Regarding, the DLS, the were re-run and the new results showed a better measurements. Also, we try to run a sample with a concentration closer to the obtained by the CMC, without success. 

Reviewer (Round 2) No discussion was added by the authors in reference to the available literature for N-oleyl cysteine derivatives.

Response: We haven´t found any N-oleyl cysteine derivatives in the literature available through our university web or physical libraries.

The conclusions have been modified by authors but there are statements not supported by the presented results.

Due to the new results obtained during the characterization, and your recommendations the conclusion were modified according to them.

Reviewer (Round 2) Language mistakes and typos are still present

More language grammar mistakes and typos were modified.

Again, thanks a lot for your suggestions!!!

Round 3

Reviewer 1 Report

Corrections have been addressed.

Author Response

Thanks a lot for your effort!

Reviewer 2 Report

The manuscript is much improved from the initial submission.

Caption Figure 6 "L-cysteine"

line 295-298 please rewrite the sentences.

Please improve the caption of Figure 8 and 9.

Figure 11 is not clear. Please indicate in the figure or in the caption what numbers on histograms and on the axis refer to.

Author Response

Changes have been made and it was proofread!

Thanks for your time and help!